# Electrophysiological Responses from the Human Tongue to the Six Taste Qualities and Their Relationships with PROP Taster Status

**DOI:** 10.3390/nu12072017

**Published:** 2020-07-07

**Authors:** Melania Melis, Giorgia Sollai, Mariano Mastinu, Danilo Pani, Piero Cosseddu, Annalisa Bonfiglio, Roberto Crnjar, Beverly J. Tepper, Iole Tomassini Barbarossa

**Affiliations:** 1Department of Biomedical Sciences, University of Cagliari, 09042 Monserrato, CA, Italy; gsollai@unica.it (G.S.); mariano.mastinu@unica.it (M.M.); crnjar@unica.it (R.C.); 2Department of Electrical and Electronic Engineering, University of Cagliari, Piazza d’Armi, I 09123 Cagliari, CA, Italy; pani@diee.unica.it (D.P.); piero.cosseddu@diee.unica.it (P.C.); annalisa@diee.unica.it (A.B.); 3Department of Food Science, School of Environmental and Biological Sciences, Rutgers University, New Brunswick, NJ 08901-8520, USA; btepper@sebs.rutgers.edu

**Keywords:** electrophysiological recording from human tongue, six taste qualities, PROP phenotype

## Abstract

Taste buds containing receptor cells that primarily detect one taste quality provide the basis for discrimination across taste qualities. The molecular receptor multiplicity and the interactions occurring between bud cells encode information about the chemical identity, nutritional value, and potential toxicity of stimuli before transmitting signals to the hindbrain. PROP (6-*n*-propylthiouracil) tasting is widely considered a marker for individual variations of taste perception, dietary preferences, and health. However, controversial data have been reported. We present measures of the peripheral gustatory system activation in response to taste qualities by electrophysiological recordings from the tongue of 39 subjects classified for PROP taster status. The waveform of the potential variation evoked depended on the taste quality of the stimulus. Direct relationships between PROP sensitivity and electrophysiological responses to taste qualities were found. The largest and fastest responses were recorded in PROP super-tasters, who had the highest papilla density, whilst smaller and slower responses were found in medium tasters and non-tasters with lower papilla densities. The intensities perceived by subjects of the three taster groups correspond to their electrophysiological responses for all stimuli except NaCl. Our results show that each taste quality can generate its own electrophysiological fingerprint on the tongue and provide direct evidence of the relationship between general taste perception and PROP phenotype.

## 1. Introduction

Taste buds are the peripheral organs of the gustatory system and are located on the surface of three different taste papillae, which are topographically arranged mostly on the tongue epithelium. These include the fungiform papillae on the anterior surface, foliate papillae on the lateral sides, and circumvallate papillae on the posterior part. Taste buds are clusters of 50–100 different types of columnar taste cells extending from the base to the top of the cluster, in which the mechanisms of taste reception and transduction are located [1]. The detection of stimuli occurs at the chemosensory apical tip of taste cells where the molecular receptors for chemicals are located. The significant redundancy and variety of the molecular receptors may reflect the importance of distinguishing nutrients from noxious substances of the environment [2].

Taste bud cells are differentiated into three types according to their ultrastructure and function: type I, type II, and type III cells [1]. Type I cells seem to have glial function, although they have been shown to present ionic currents implicated in salt taste transduction [3]. Type II cells, named “receptor cells”, express G protein-coupled receptors (GPCRs) and their downstream effectors for the transduction of sugar, amino acid, and/or bitter stimuli [1]. Some type II cells express members of the family of taste receptor type II (T2R) which are activated by bitter-tasting compounds [4,5], while others express heterodimeric T1R family receptors, which are excited by sweet- or umami-tasting stimuli [6,7,8]. Type II cells are most likely involved in the transduction mechanism of fat, which is detected by at least two types of membrane proteins (GPR120 and the multifunctional CD36 scavenger receptor) [9]. Stimulation of type II cell molecular receptors by specific stimuli activates the following molecular/electrophysiological mechanisms: release of G protein (Gβγ) dimers, stimulation of phospholipase Cβ2, mobilization of intracellular Ca^2+^ [10,11], opening of transient receptor potential cation channel subfamily M member 5 (TRPM5), and depolarization of taste cells [12,13,14]. These cells do not form specialized synapses with their closely neighboring sensory nerve fibers but communicate with them via non-vesicular transmitter release. Type III cells are named “presynaptic cells” because they have ultrastructurally recognizable synaptic membranes. However, they also contain the needed machinery to detect sour stimuli and generate a depolarizing current [15,16,17,18,19].

The specific sensitivities of taste cells within taste buds provide the basis for discrimination across taste qualities [20]. The sensitivity and selectivity of taste bud cells have been studied by Ca^2+^ imaging and electrophysiological experiments [17,21]. These results indicate that taste buds contain several taste cells that respond to one specific taste quality. Nevertheless, buds also contain taste cells that can detect multiple taste qualities [17,21,22] and some taste bud cells express more than one type of receptor. This multiple responsiveness of taste bud cells may reflect the information processing that takes place within taste buds via cell-to-cell communications and modulation [1,2,20,23,24,25,26]. Specifically, types II and III cells within a taste bud communicate with one another with divergence and convergence of signals, thus, enabling taste buds to encode chemical identity, nutritional value, and concentration of taste stimuli before transmitting signals to the central neurons [20]. The signals integrated in taste buds are transmitted, through taste fibers of the VII, IX, and X cranial nerves, to the solitary tract nucleus of the medulla, thalamus, and hence to the cortex gustatory areas where the sensation is consciously recognized [27]. A multitude of factors influence this sensation mixing with other signals, such as physiological state (hunger/fullness), emotional state, attention to the taste task, etc. [28,29].

In humans, taste perception varies greatly among individuals and is one of the most significant determinants influencing food preferences and therefore eating behavior, metabolism, and health [27,30]. Within this nutritional context, the genetic ability to taste the bitter compound 6-*n*-propylthiouracil (PROP) has gained a consistent and appreciable consideration as a marker of general taste perception, dietary preferences, and habits that can impact on nutrition and health of individuals [30]. This important role assigned to PROP tasting is based on findings reporting that subjects who perceive PROP as intensively bitter (PROP super-tasters), have a higher sensitivity and lower preference, than non-tasters, to various oral stimuli, including other bitter substances [31,32,33,34,35,36], sweet stimuli [37], sour compounds [38], umami taste [39], irritants [40,41], high-fat/high-energy foods [42,43,44], astringent substances [45], and fruits and vegetables [46,47,48]. Some authors suggested that PROP-related sensory variations may be associated with olfactory function [49,50], and that PROP tasting may affect the perception of foods via aromas or flavors [51,52]. Data have also shown relationships between PROP phenotype or genotype and longevity [53] or health parameters including: antioxidant status [54], body mass index (BMI) [55,56], metabolic changes that impact on body mass composition [57,58], smoking behaviors [59], consumption of alcoholic beverages [41], respiratory infection and rhinosinusitis [60,61,62,63,64,65,66,67], taste disorders [68], and development of colonic neoplasm [69,70,71], and even neurodegenerative diseases [72]. However, the validity of PROP tasting as a biomarker has been questioned by other authors who have reported results not confirming these associations [73,74,75,76,77,78]. Divergent results may be due to the fact that sensory analyses are carried out by psychophysical testing procedures, which are based on self-reports and therefore can produce highly subjective evaluations. Our laboratory has adopted a simple and reliable technique that permits the collection of quantitative measures of peripheral taste function, by electrophysiological recordings from the human tongue, as objective data that are not influenced by the individual’s subjective biases [79,80,81]. We found differences in the bioelectric activity from the tongue in response to PROP that were consistent with subjects’ PROP genotype and phenotype [79]. Furthermore, electrophysiological responses to oleic acid were consistent with variation in the *rs1761667* SNP of the gene coding for the fatty acid scavenger receptor CD36 [81]. In addition, we developed a method capable of automatically discriminating among subjects belonging to three PROP taster categories (super-tasters, medium tasters, and non-tasters), by exploiting features extracted from electrophysiological recordings [80]. A low-cost handheld tool for the acquisition of the signal was also developed [82].

The purpose of the present work is twofold. Firstly, we extend our previous results [79,80,81] to characterize the bioelectrical activity from the human tongue in response to the six taste qualities. By studying all six taste qualities, we obtained important information about the organization and function of the human peripheral taste system. Importantly, our non-verbal method of taste recordings from the tongue is not influenced by the many factors that affect verbal self-reports of taste signals that are interpreted at the level of the cortex.

Secondly, we verified the validity of using PROP tasting as a biomarker of general taste perception by comparing the electrophysiological response to each taste quality in subjects belonging to the three PROP taster categories (super-taster, medium taster, and non-taster), as an objective measure of their taste sensitivity for the different taste qualities. Since our previous studies showed robust relationships between the magnitude of electrophysiological taste responses and fungiform papillae densities, the latter measurements were also collected here.

## 2. Materials and Methods

### 2.1. Subjects

Thirty-nine Caucasian non-smoking subjects (11 males, 28 females, age 28.31 ± 1.03 years) were recruited through usual procedures at the University of Cagliari. They were originally from Sardinia, Italy. No statistical methods were done to predetermine sample size. However the following guiding criteria were used: sample size was similar to the one already employed in electrophysiological experiments that assess the degree of activation of the receptors under study [79,81]; subjects were recruited to form three similar equal-sized PROP-taster groups, matched for gender and age and had at least 10 subjects per group. Subjects were normal weight with body mass index (BMI) ranging from 20.2 to 24.8 kg/m^2^. None had food allergies or were following a specific diet or taking medications that could interfere with taste sensitivity. Their taste function for the four basic tastes was screened by the taste strip test (Burghart Messtechnik, Wedel, Germany) to exclude any taste impairment. Subjects read and signed an informed consent form. This study was carried out in accordance with the latest revision of the Helsinki Declaration, and all procedures have been approved by the Ethical Committee of the University Hospital Company (AOU) of Cagliari, Italy. The trial was registered at ClinicalTrials.gov (identifier number is UNICADBSITB-1).

### 2.2. Experimental Procedure

Each subject was tested on two consecutive days: on the first day, he/she was classified for PROP taster status. On the second day, he/she was tested for the electrophysiological responses to stimulations with six taste qualities and for density of fungiform papillae. All subjects had to abstain from drinking (except water), eating, using oral care products, or chewing gum for at least 2 h prior to testing. They had to be in the experiment room 15 min before the starting of the test (9.00 a.m.) in order to acclimate to the constant environmental conditions (40–50% relative humidity; 23–24 °C). Women were examined around the sixth day of the menstrual cycle to avoid changes of taste function resulting from the estrogen phase [83]. Stimuli were presented at room temperature as solutions in spring water, which were prepared 1–2 days before each session and stored in a refrigerator until 1 h before testing. Each subject was tested for each taste quality in a double-blinded and counterbalanced order. The interstimulus interval was set at 1 h. At the end of the electrophysiological recording in response to each taste quality, the subject, who was instructed to rate only the chemosensory perception, scored the perceived intensity by placing a mark on the Labeled Magnitude Scale (LMS) [84]. The LMS is a 100-mm scale (semi-logarithmic) in which seven labeled verbal descriptors (barely detectable, weak, moderate, strong, very strong, strongest imaginable) are arranged along the length of the scale. The LMS gives subjects the freedom to evaluate the perceived taste intensity for a taste stimulus relative to the strongest imaginable oral stimulus ever experienced in life.

### 2.3. PROP Taster Status

Subjects were classified for their PROP taster status by two scaling methods. All were first assessed using the three-solution test according to Tepper et al. 2001 [85], which has been validated in numerous studies [39,86,87,88,89]. The test consists of the perceived taste intensity ratings of three suprathreshold sodium chloride (NaCl; 0.01, 0.1, 1.0 mol/L) (Sigma-Aldrich, Milan, Italy) and PROP (0.032, 0.32, and 3.2 mmol/L) (Sigma-Aldrich) solutions, which were collected by using the LMS [84]. Concentrations (10 mL samples) were presented in a random order. Subjects who gave lower ratings to PROP than to NaCl were classified as PROP non-tasters, those who gave overlapping ratings to the two chemicals were classified as medium tasters, and those who gave higher ratings to PROP than to NaCl were classified as super-tasters. After a 1-h period, subjects were classified as belonging to a PROP taster group (super-taster, medium taster, or non-taster) by using the impregnated paper screening test [90,91]. With this method, the PROP and NaCl were presented to each subject by placing two paper disks impregnated with solutions of two stimuli (PROP, 50 mmol/L and NaCl, 1.0 mol/L) on the tip of the tongue for 30 s. Additionally, this method used the LMS to rate the perceived taste intensity. Subjects who rated the PROP higher than 67 were classified as super-tasters, those who rated the PROP lower than 15 mm on the scale were classified as non-tasters, all others were categorized as medium tasters [91]. Only subjects who were categorized in a similar way by the two PROP screening methods were included in the study. Based on the classification, which was documented by three-way ANOVA, 14 subjects were classified as non-tasters (35.89%), 15 were medium tasters (38.46%), and 10 were super-tasters (25.64%) (Appendix A). The basic anthropometrics of the three PROP taster groups are summarized in Table 1.

### 2.4. Electrophysiological Recordings from the Tongue

Differential electrophysiological recordings from the tongue of subjects were performed according to Sollai et al. [79,81]. Briefly, recordings were performed between two silver electrodes. The first electrode was a silver wire (50 mm) that had the distal end rolled up to form a ball (dia.: about 5 mm) to obtain a good electrical contact when it was positioned in contact with the ventral surface of the tongue. The second electrode (patent WO 2017/212377) consisted of a silver film (100 nm thick) deposited, by means of evaporation in high vacuum, on a very thin (13 μm) polyimide layer (Kapton©, DuPont, Wilmington, DE, USA). This electrode was positioned in perfect adhesion with the left side of the tip of the tongue’s dorsal surface taking advantage of its extreme thinness. Its distal part had a circular hole (6 mm diameter), which leaves uncovered a small area of the left side of the tip of the tongue surface. This area was the zone where taste stimulations were delivered during the electrophysiological recordings, and the density of fungiform papillae were calculated as described below. The dorsal electrode was isolated by a film of biocompatible material (Parylene C, 2 µM thick) excluding the area that must be in electrical contact with the tongue to detect the electrophysiological signal. The ground terminal of the measuring system was connected to a third disposable adhesive electrode placed in an electrically neutral position (CDES003545, SpesMedica, Genova, Italy). Figure 1 shows the two electrodes used for the differential electrophysiological recordings and how they were positioned in contact with the human tongue. The bio-potentials detected by the electrodes were recorded by a high-input impedance polygraph (for human use, Porti7 portable physiological measurement system; TMS International B.V., Zevenhuizen, The Netherlands), which is an isolated, certified Class IIa medical device. After achieving a stable baseline, signals were digitized, recorded, and visualized in real time on a PC by PolyBench software (TMS International B.V., Zevenhuizen, The Netherlands). For each taste quality, the recording lasted 55 s (20 s baseline, 15 s during taste simulation, and 20 s after stimulation, i.e., after the paper disk was removed). 

The waveform of bio-potentials was analyzed by Clampfit 10.0 software (Molecular Devices, Sunnyvale, CA, USA). The voltage changes (amplitude values) in response to six taste qualities were obtained by subtracting the baseline value from the voltage at 0.1, 2.5, 5, 10, and 15 s from stimulation onset. The rate of potential variation (mV/s) was also calculated at the same time intervals.

### 2.5. Taste Stimulations

Taste stimulations were delivered by placing, for 15 s a paper disk (6 mm dia.) impregnated with 30 µL of taste solutions, or 30 µL of undiluted oleic acid, on the circular area of the tongue surface, which was left free by the hole of the second electrode. Sucrose (200 mM), NaCl (200 mM), citric acid (5.2 mM), caffeine (10 mM), and monosodium glutamate (MSG) (160 mM) solutions were used to represent the five primary taste qualities (sweet, sour, salty, bitter, and umami). The concentration for each stimulus was chosen based on preliminary tests. 

In order to verify that the potential variations recorded were certainly due to taste stimulations and not mechanical stimulations caused by placing the disks, subjects were also tested with dry paper disks.

### 2.6. Density Assessments of Fungiform Taste Papillae

The total number of fungiform papillae was measured in the same circular area of the tongue where the stimulations were performed during electrophysiological recordings according to Melis et al. 2013 [92]. This area was located in the left side of the anterior surface of the tip of the tongue, closest to the midline, and spaced 4 mm from the edge of the tongue according to our previous works [79,81]. This shift with respect to the area that provides reliable measurements of papilla density in high correlation with the total number on the tongue [93] was indispensable to keep the area perfectly overlapping with the area left uncovered by the second electrode. This area was stained by using a blue food dye (E133, Modecor Italiana, Cuvio, Italy), and then photographs were taken using a Canon EOS D400 (10 megapixels) camera with a lens (model: EF-S 55–250 mm). The digital images were analyzed using the “zoom” option in the Adobe Photoshop 7.0 program. The fungiform papillae were separately picked out and counted by three expert operators who were not informed about the PROP taster status of subjects [79,92,93]. The density/cm^2^ was calculated.

### 2.7. Statistical Analysis

Repeated-measures ANOVA was used to analyze differences of mean values ± SEM of the bio-signal amplitude (mV) and potential variation rate (mV/s) evoked by the six taste qualities at 2.5, 5, 10, and 15 s after the application of stimulation. Data were also analyzed across PROP taster groups. Two-way ANOVA was used to compare differences in perceived intensity rating for the six taste qualities in super-tasters, medium tasters, and non-tasters. One-way ANOVA was used to compare differences in density of fungiform papillae according PROP taster status. Data were verified for the assumptions of normality, sphericity (when applicable), and homogeneity of variance. A Greenhouse–Geisser correction or Huynh–Feldt correction was applied whenever the sphericity assumption was violated. Post hoc comparisons were conducted with the Fisher’s least significant difference (LDS) test or Duncan’s test when the assumption of homogeneity of variance was violated. Statistical analyses were conducted using STATISTICA for WINDOWS (version 7; StatSoft Inc., Tulsa, OK, USA). *p* values < 0.05 were considered significant.

## 3. Results

### 3.1. Electrophysiolgical Responses to Taste Stimulation with Six Taste Qualities

The electrophysiological recordings from the human tongue allowed the measurement of monophasic bioelectrical potential changes in response to taste stimulations, with respect to the baseline. NaCl stimulation evoked depolarizing monophasic potential changes, whereas all the other stimuli evoked hyperpolarizing monophasic potential changes. However, the waveform analysis of bioelectrical potentials allowed the observation that each stimulus determined a characteristic time course of the potential variation during stimulation. Examples of this are shown in Figure 2. Stimulations with dry paper disks produced no potential changes. 

The mean values ± SEM of the potential amplitude (mV) and of potential change rate (mV/s) determined after 0.1, 2.5, 5, 10, and 15 s from the stimulus application in response to the six taste qualities are shown in Figure 3. Repeated-measures ANOVA revealed that the time course of the amplitude and rate of potential change during stimulation time depended on taste quality (amplitude: F_20,820_ = 37.840; *p* < 0.0001 and rate: F_20,820_ = 5.5235; *p* < 0.0001). Post hoc comparison showed that all stimuli produced significant potential changes, with respect to baseline, after 2.5 s from stimulus application (*p* ≤ 0.0016; Fisher LDS). The largest positive potential changes were recorded in response to citric acid and sucrose (*p* < 0.0001; Fisher LDS), the intermediate ones in response to caffeine and MSG (*p* < 0.0001; Fisher LDS), and the lowest in response to oleic acid (*p* = 0.0016; Fisher LDS); negative potential changes were recorded in response to NaCl (*p* < 0.0001; Fisher LDS). Successively, the voltage variations elicited by citric acid decreased significantly at 10 s (*p* = 0.0388; Fisher LDS); those in response to sucrose, caffeine, and MSG did not change during the stimulation time (*p* > 0.05); while that in response to oleic acid slowly continued to increase (significantly at 10 s; *p* = 0.0198; Fisher LDS) to the end of stimulation. The depolarization in response to NaCl did not change during the stimulation time (*p* > 0.05).

The quickest potential changes were recorded at 0.1 s in response to citric acid (*p* < 0.0001; Fisher LDS), then the values rapidly decreased across the time course of electrophysiological recordings, being halved at 2.5 s (*p* < 0.0001; Fisher LDS) and halved again at 5 s (*p* = 0.0011; Fisher LDS). Slower potential changes were determined at 0.1 s in response to sucrose (*p* < 0.0001; Fisher LDS), which significantly decreased only at 5 s (*p* ≤ 0.0047; Fisher). NaCl and caffeine produced intermediate values of potential change rate, which did not vary with time (*p* ≤ 0.021; Fisher LDS). The lowest values, unchanged over time, were observed with MSG and oleic acid (*p* ≤ 0.026; Fisher LDS).

### 3.2. Relationship between Electrophysiolgical Responses to Taste Stimulations with Six Taste Qualities and PROP Taster Status

The mean values ± SEM of the potential amplitude (mV) and of potential change rate (mV/s) determined after 0.1, 2.5, 5, 10, and 15 s from the application of stimulation with the six taste qualities according to PROP taster status are shown in Figure 4. Repeated-measures ANOVA revealed that the time course of the amplitude of bioelectrical signals during stimulation time, depended on taste quality and PROP taster status of subjects (F_40,772_ = 2.1256; *p* < 0.0001). After 2.5 s from stimulation, all stimuli evoked significant potential changes in all subjects (*p* ≤ 0.00013; Fisher LDS), except for oleic acid in the non-taster group (*p* > 0.05). PROP super-tasters showed larger amplitudes than other PROP taster groups. Specifically, super-tasters showed larger signal amplitudes in response to NaCl and sucrose compared to those of other PROP taster groups at 2.5, 5, 10, and 15 s (*p* ≤ 0.042; Fisher LDS), as well as larger signal amplitudes in response to MGS and citric acid than those of other taster groups at 2.5 and 5 s (*p* ≤ 0.038; Fisher LDS). Super-tasters also showed larger responses to oleic acid than those of non-tasters at 10 and 15 s (*p* ≤ 0.019; Fisher LDS) and medium tasters showed higher values for oleic acid relative to non-tasters at 15 s (*p* = 0.036; Fisher LDS). The amplitude of signals in response to caffeine was slightly higher in super-tasters than in other PROP taster groups but not in a significant way.

The mean values ± SEM of the potential change rate (mV/s) determined after 0.1, 2.5, 5, 10, and 15 s from the application of stimulation in response to the six taste qualities according to PROP taster status are shown in Figure 5. Repeated-measures ANOVA revealed that the time course of the rate of bioelectrical signals during stimulation time, depended on taste quality and PROP taster status of subjects (F_40,772_ = 5.541; *p* < 0.00001). PROP super-tasters showed a quicker hyperpolarization with respect to the other taster groups, at 0.1 and 2.5 s, in response to sucrose and citric acid (*p* ≤ 0.0427; Fisher LDS); this same effect was only seen for caffeine at 0.1 (*p* ≤ 0.0239; Fisher LDS).

### 3.3. Relationship between Rating of the Perceived Intensity for the Six Taste Qualities and PROP Taster Status

The mean values ± SEM of the rating of the perceived intensity for the six taste qualities according to PROP taster status are shown in Figure 6. PROP super-tasters gave statistically significant higher intensity ratings to all stimuli with respect to the other taster groups (*p* ≤ 0.019; Fisher LDS, subsequent to two-way ANOVA), except for oleic acid for which the ratings of super-tasters were higher than those of non-tasters only (*p* = 0.0436; Fisher LDS, subsequent to two-way ANOVA). No significant difference related to PROP taster status was found for NaCl.

### 3.4. Relationship between Fungiform Papillae Density and PROP Taster Status 

The mean values ± SEM of density of fungiform papillae determined in super-tasters, medium tasters, and non-tasters are shown in Figure 7. One-way ANOVA showed that the density of fungiform papillae varies with PROP taster status (F_2,33_ = 13.105; *p* = 0.00006). PROP super-tasters had a higher density than medium tasters (*p* = 0.021; Fisher LDS test), who showed higher values than non-tasters (*p* = 0.0072; Fisher LDS test).

## 4. Discussion

The first aim of this work was to characterize the electrophysiological responses evoked in taste buds on a localized area of the human tongue by stimulation with the six taste qualities, as a direct and quantitative measure of the degree of activation of the peripheral taste system generated by each taste stimulus. Interestingly, we found that each taste quality evoked a monophasic voltage change with a characteristic time course during stimulation. In fact, a comparative analysis of values of potential amplitude and of potential change rate, determined during stimulation, revealed that the waveform of the signal depends on the taste quality of stimulus. Although, the bioelectrical activity generated by all stimuli was represented by a monophasic potential change characterized by a fast-initial variation followed by a slow decline, NaCl was the only stimulus that evoked negative potential variations, while all other stimuli evoked positive potential changes, which nevertheless were differed from one another. 

Differences in type of taste cells activated and in transduction mechanisms across taste qualities might explain these differences. Sweet, bitter, umami, and fat stimuli directly activate type II cells, which express specific receptors and, via ATP released during their tastant-induced stimulation, indirectly activate the adjacent type III cells, which instead directly respond to sour stimuli [17,21]. As mentioned above, our results showed that sucrose, citric acid, caffeine, MSG, and oleic acid evoked specific positive potential changes. Sucrose and citric acid generated the largest hyperpolarization mostly in the first part of the recordings. However, the responses to these two stimuli differed in their last portion; for citric acid, signal amplitude decreased significantly after 10 s, while for sucrose it did not change until the end of the stimulation. The responses to these two stimuli also differed for the hyperpolarization rate. For citric acid, the hyperpolarization arose very quickly, while for sucrose it was slower, with values that were half those evoked by citric acid. Caffeine and MSG evoked signals with intermediate amplitudes in the first part of the recordings, but afterward the signal amplitude distinguished the response to these two stimuli. Differences in the hyperpolarization rate also allowed us to distinguish the electrophysiological responses to these two stimuli; intermediate values were found in response to caffeine, while very slow signals were recorded in response to MSG. Finally, we recorded the lowest and slowest hyperpolarization in the responses to oleic acid, in which the signal amplitude reached values comparable to those of other stimuli only at the end of stimulation. This slow electrophysiological activation in response to oleic acid, which has been already observed in our previous work [81], could depend on the high surface tension of this molecule that determines its slow diffusion toward the cell that detects it. The steep-onset hyperpolarization evoked by citric acid (which was particularly evident in super-tasters who have a high density of papillae), compared to that elicited by other stimuli (sucrose, caffeine MGS, oleic acid) may be explained by differences in the transduction mechanisms across stimuli [1]. Organic acids permeate through the membrane, acidify the cytoplasm and intracellular H^+^, by blocking a proton-sensitive K^+^ channel, and depolarize the cell membrane. On the other hand, the transduction mechanisms of sucrose, caffeine, MGS, or oleic acid are mediated by GPCRs, causing activation and diffusion of second messengers, which need more time to depolarize the cell. 

As noted above, the negative potentials we observed in response to NaCl stand in contrast to those of all other taste stimuli (which showed positive potentials). Although the taste bud cells involved in the salty taste transduction have not been conclusively identified [1,2], salty is the only taste quality known to be transduced by some Type I cells, which may exhibit depolarizing ionic currents due to direct Na^+^ permeation through membrane ion channels [3]. However, we cannot exclude that the salty stimulus may undergo an ionic dissociation directly recorded by the measuring electrode, thus superposing with the bio-signal itself. Future studies will cast a light on this issue. 

Taken together, these results indicate that each taste quality can generate a specific electrophysiological response in taste buds, and they strongly support other data showing that cell-to-cell signal interactions within taste buds, may represent a primary localization step where taste quality discrimination begins [20]. 

Our results shed additional light on the validity of using the PROP phenotype as a biomarker of general taste perception. We found a direct relationship between PROP sensitivity and the electrophysiological response evoked by the six taste qualities in the buds of a localized area of human tongue. Specifically, the largest and quickest responses were recorded in PROP super-taster subjects, who had the highest density of fungiform papillae in the same area of the tongue where stimulations were applied during the recordings. Smaller and slower responses were observed in medium taster and non-tasters, who had lower densities of papillae in the same area of the tongue. The analysis of the time course of parameters defining the waveform of signals evoked by NaCl, sucrose, caffeine, MGS, and citric acid revealed that the super-taster phenotype, which was associated with a highest density of papillae, confers additional advantage already in the initial phase (2.5 s) of the response, which is certainly the most significant one to induce a behavioral response. Super-tasters showed a more prompt and intense response to sucrose and citric acid, a more intense one to NaCl and MGS, and a more rapid one to caffeine, as compared to medium tasters and non-tasters. Differently, the super-taster phenotype was important to elicit a large potential change in the last part of the response to oleic acid. The signal amplitude values slowly increased during stimulation in super-tasters, while no changes in the whole time course of recordings were found in medium tasters and non-tasters. This extended activation evoked by oleic acid in super-tasters, having a higher density of papillae, may reflect the high surface tension of the molecule that determines its slow diffusion toward the taste bud cells. 

These results strongly support previous psychophysical data showing a direct relationship between perception of a wide range of oral stimuli and PROP taster status [31,32,33,34,35,36,37,38,42,43,44,45,46,47,48], that can be linked to changes in density of papillae across the PROP taster groups [37,92,93,94,95,96]. This suggests that the phenotypic expression of the trait, which is strongly associated with density of papillae, is a critical determinant of the electrophysiological responses to the six taste qualities. Furthermore, the changes found among the PROP taster groups, which differed in papilla density, confirm that the monophasic potentials recorded effectively represent the summated response of stimulated taste cells, as already shown in our previous studies [79,81] and also in other sensory organs of vertebrates, including the human olfactory epithelium [97,98,99]. It is important to acknowledge, however, that some studies report weak or no associations between PROP taster status and papillae density [100,101]. As pointed out by Tepper et al. [102] and Dinnella et al. [101], personal factors such as age, gender, smoking, bodyweight, and modified genes can influence papillae density as well as PROP tasting, potentially undermining interrelationships between PROP, taste perceptions, and papillae density. Moreover, the differences in the screening procedures can cause inconsistencies among studies [102]. This topic deserves further investigation.

Importantly, the differences in the electrophysiological responses to sucrose, caffeine, MGS, oleic acid, and citric acid that we recorded in the PROP taster groups agree with the intensity ratings given by these subjects during oral stimulations, indicating that our bioelectrical measurements are consistent with common human psychophysical observations. It is noteworthy that the amplitude of electrophysiological response to NaCl by super-tasters was larger than those for medium tasters and non-tasters, however, the perceived intensities reported by subjects did not vary with PROP taster status. This is consistent with the psychophysical procedures for the classification of subjects by PROP taster status, which use NaCl as standard control showing that the rating of perceived intensity for this stimulus does not vary with the PROP phenotype of subjects [85]. It is also worth highlighting that we found lack of agreement between signal amplitude and perceived intensity for caffeine, relatively to PROP taster groups. The perceived intensities for caffeine reported by super-tasters were higher than those for medium tasters and non-tasters, while the response amplitude did not vary with PROP taster status, though signals for super-tasters were faster. This result may seem inconsistent with data showing that the perceived intensity of bitterness is associated with signal amplitude when using PROP [79]. Although caffeine and PROP are both bitter-taste stimuli, they are known to activate different TAS2Rs receptors [103]. 

## 5. Conclusions 

The findings of the present study were achieved thanks to a novel electrophysiological recording technique from the human tongue to directly and quantitatively measure the degree of activation of the peripheral gustatory system in response to taste stimuli [79,81]. Results show that each taste quality can elicit a specific and characteristic electrophysiological response in taste buds, consistent with our current understanding of the biological mechanisms of taste transduction and cell-to-cell communications [1,2,20,23,24,25,26]. 

In addition, our data provide the first direct and objective demonstration of the role of PROP phenotype in individual variability of general taste perception. They show that the influence of PROP status on the amplitude of electrophysiological response may reflect a summation effect associated with differences in papillae density, a well-known anatomical characteristic of the PROP phenotype [37,92,93,94,95,96].

Finally, psychophysical measures have played a critical role in understanding human chemosensory experiences, by permitting the determination of taste responses at the CNS level, which can influence food choices and eating behaviors. Nevertheless, because taste intensity measures are subjective and sensitive to reporting bias, they are less useful for gaining insights into taste mechanisms, particularly those rapid responses that occur within the initial few seconds of oral exposure to a stimulus. The present work helps to fill gaps in knowledge of these processes by combining traditional behavioral methods with our electrophysiological recording method. 

## Figures and Tables

**Figure 1 nutrients-12-02017-f001:**
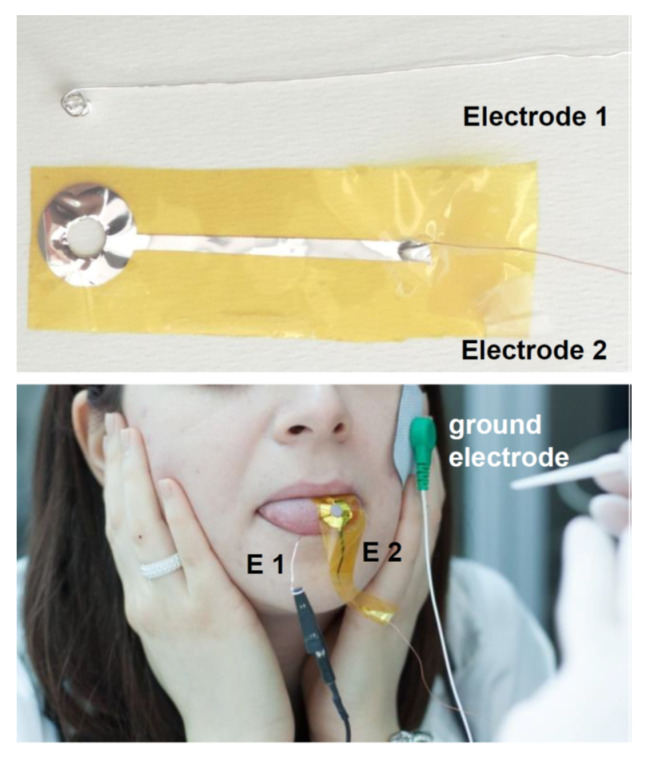
Photographs showing the two electrodes for the differentials electrophysiological recordings and how they were positioned in contact with the human tongue.

**Figure 2 nutrients-12-02017-f002:**
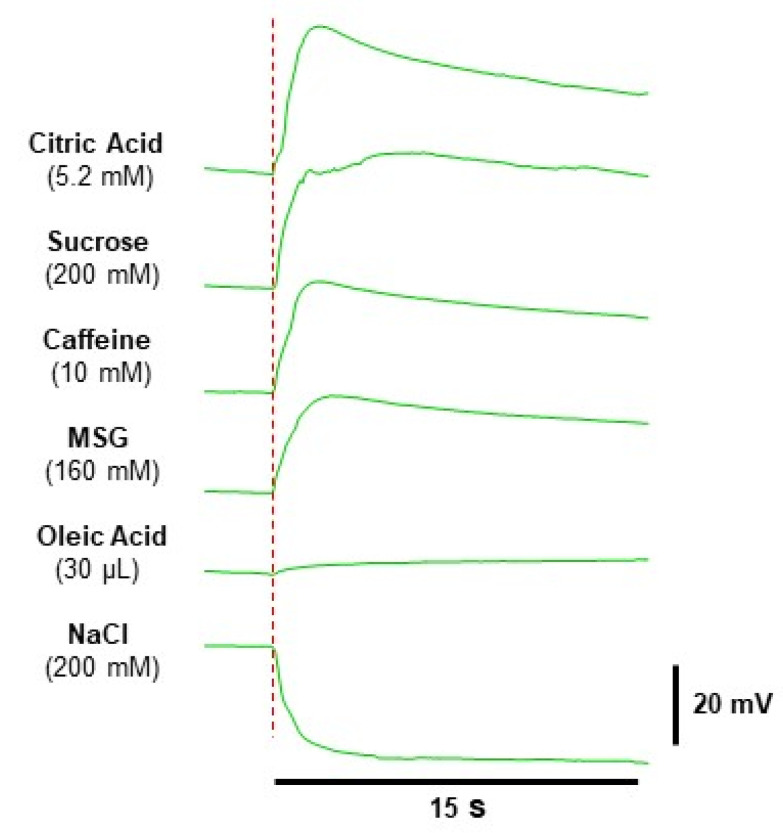
Examples of electrophysiological recordings in a super-taster subject in response to 30 µL of citric acid, sucrose, NaCl, caffeine, and monosodium glutamate (MSG) solutions or oleic acid. The very first data point on the left side of the electrophysiological recordings represents the baseline.

**Figure 3 nutrients-12-02017-f003:**
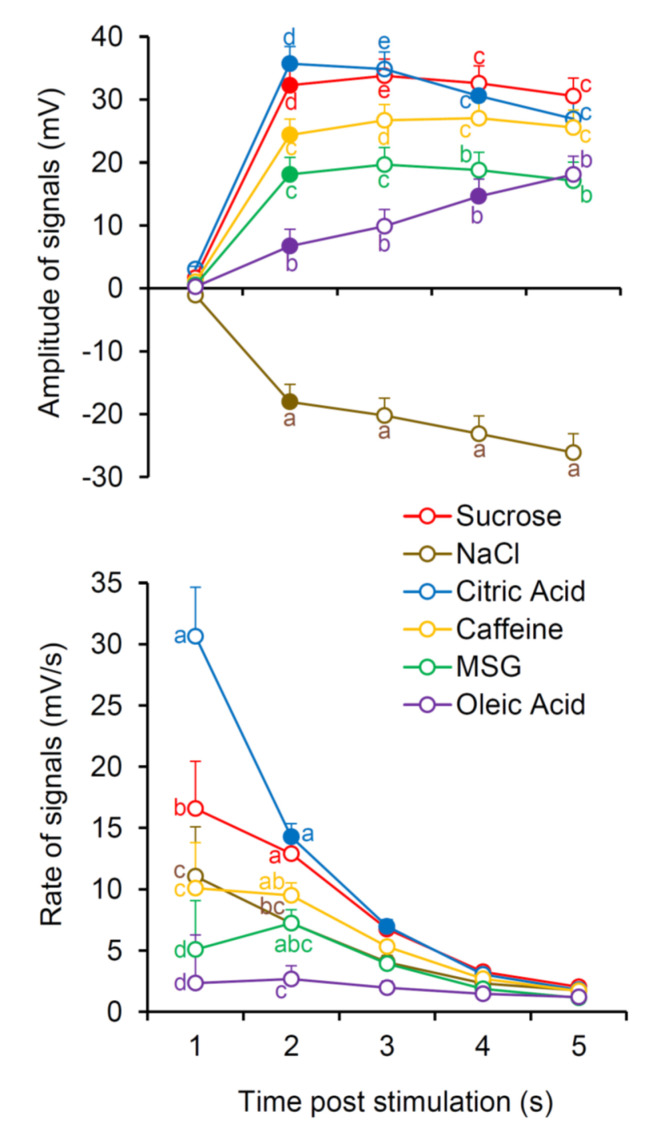
Time course of potential amplitude (mV) and of potential change rate (mV/s) of bioelectrical signals in response to taste stimulation with six taste qualities. Data (mean values ± SEM) determined after 0.1, 2.5, 5, 10, and 15 s after application of taste stimulation are shown. Numbers 1, 2, 3, 4, 5 on the *X*-axis correspond to 0.1, 2.5, 5, 10, 15 s after stimulation, respectively. *n* = 39. Solid symbol indicates significant difference with respect to the previous value of the corresponding group (*p* ≤ 0.0016; Fisher least significant difference (LDS) subsequent to repeated-measures ANOVA). Different letters indicate a significant difference with respects to another stimulus in the corresponding time (*p* ≤ 0.041; Fisher LDS, subsequent to repeated-measures ANOVA).

**Figure 4 nutrients-12-02017-f004:**
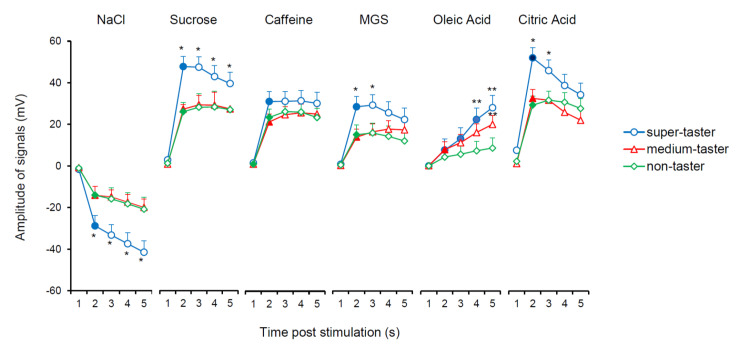
Time course of potential amplitude (mV) of bioelectrical signals in response to taste stimulation with six taste qualities according to PROP taster status. Data (mean values ± SEM) determined after 0.1, 2.5, 5, 10, and 15 s after application of taste stimulation. Numbers 1, 2, 3, 4, 5 on the *X*-axis correspond to 0.1, 2.5, 5, 10, 15 s after stimulation, respectively). *n* = 10 super-tasters, *n* = 15 medium tasters, and *n* = 14 non-tasters. Solid symbol indicates significant difference with respect to the previous value of the corresponding group (*p* ≤ 0.011; Fisher LDS, subsequent to repeated-measures ANOVA). * Indicates a significant difference with respects to the corresponding values of other taster groups (*p* ≤ 0.042; Fisher LDS, subsequent to repeated-measures ANOVA). ** Indicates a significant difference with respects to the corresponding values of non-tasters (*p* ≤ 0.036; Fisher LDS or Duncan’s test, subsequent to repeated-measures ANOVA).

**Figure 5 nutrients-12-02017-f005:**
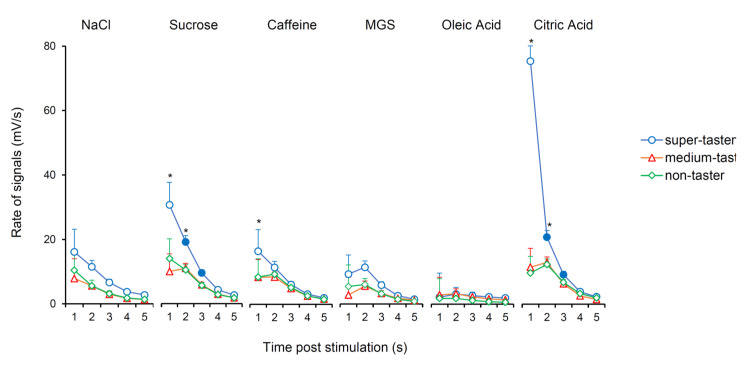
Time course of potential change rate (mV/s) of bioelectrical signals in response to taste stimulation with six taste qualities according to PROP taster status. Data (mean values ± SEM) determined after 0.1, 2.5, 5, 10, and 15 s after application of taste stimulation. *n* = 10 super-tasters, *n* = 15 medium tasters, and *n* = 14 non-tasters. Numbers 1, 2, 3, 4, 5 on the *X*-axis correspond to 0.1, 2.5, 5, 10, 15 s after stimulation, respectively. Solid symbol indicates significant difference with respect to the previous value of the corresponding group (*p* ≤ 0.0099; Fisher LDS, subsequent to repeated-measures ANOVA). * Indicates a significant difference with respects to the corresponding values of other taster groups (*p* ≤ 0.0427; Fisher LDS, subsequent to repeated-measures ANOVA).

**Figure 6 nutrients-12-02017-f006:**
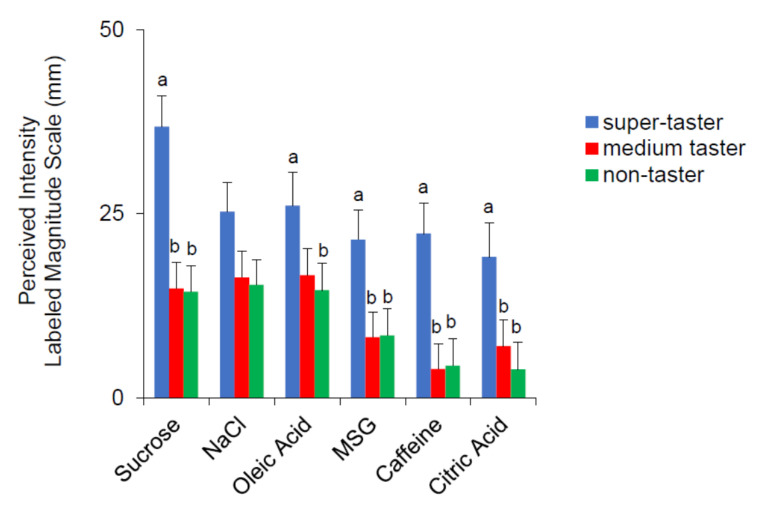
Perceived intensity rating for the six taste qualities in super-tasters (*n* = 10), medium tasters (*n* = 15), and non-tasters (*n* = 14). Different letters indicate a significant difference (*p* ≤ 0.0436; Fisher LDS, subsequent to repeated-measures ANOVA).

**Figure 7 nutrients-12-02017-f007:**
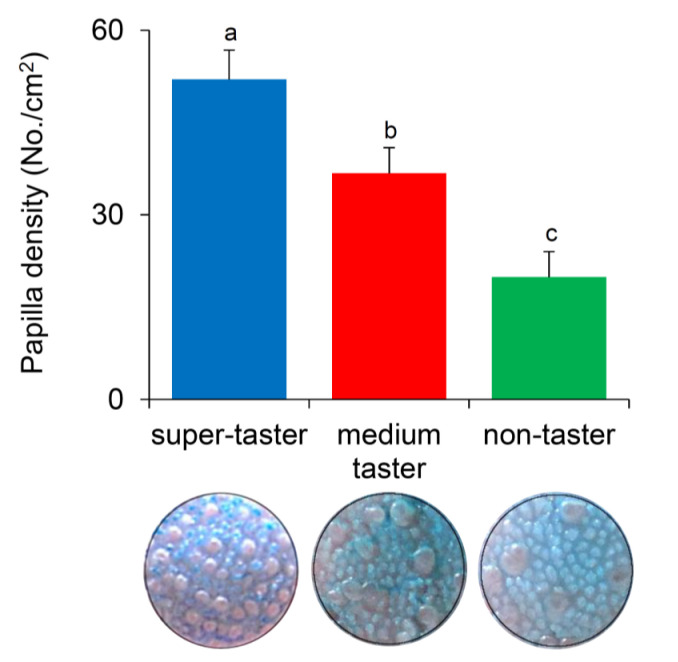
Density of fungiform papillae in super-tasters (*n* = 10), medium tasters (*n* = 15), and non-tasters (*n* = 14). Different letters indicate a significant difference (*p* ≤ 0.021; Fisher LDS test subsequent one-way ANOVA). Images of the stained area (6 mm dia.) of the tongue where the fungiform papillae were counted in a representative super-taster, medium taster, and non-taster are also shown.

**Table 1 nutrients-12-02017-t001:** Basic anthropometric features of three 6-*n*-propylthiouracil (PROP) taster groups.

	Super-Taster (*n* = 10)	Medium Taster (*n* = 15)	Non-Taster (*n* = 14)
Age (years)	27.2 ± 2.07	28.87 ± 1.87	28.43 ± 1.75
Female/Male	7/3	9/6	12/2
BMI (kg/m^2^)	21.20 ± 0.72	22.02 ± 0.58	21.49 ± 0.61

Values are means ± SEM. BMI, body mass index.

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
