# Peer review of "Electrophysiological Responses from the Human Tongue to the Six Taste Qualities and Their Relationships with PROP Taster Status"

_nutrients, 2020, doi:10.3390/nu12072017_

Round 1

Reviewer 1 Report

Although multiple articles have investigated the relationship between PROP taster status, fungiform papilla density and taste sensitivity in humans, the existence of a correlation remains controversial. The article by Melis and colleagues presents new data that support a correlation between PROP taster status, fungiform papilla density and taste sensitivity. While the study could have been even stronger with a larger sample size, the data presented here are convincing as the methodologies are very thorough and include two different measures of taste sensitivity: taste intensity rating by the subjects on a Labelled Magnitude Scale, and unbiased electrophysiological recording. In addition to the five basic taste modalities, the authors also included fat taste sensitivity, which has been proposed to be the sixth basic taste, for a broader picture. The article is well written and includes very detailed procedures and statistical analysis.

Only a few minor comments arose from reading the manuscript:

  1. Line 186: what was the oleic acid concentration? What was the vehicle?
  2. Line 193: dry paper disks were used as controls, but it is unclear how the generated control data were included in the analysis. Please clarify. Also, were some paper disks impregnated with water only, or with a texture agent only, as vehicle control (for sweet, bitter, sour, umami, salty compounds) and control for the texture/mouthfeel of oleic acid, respectively? Were the subjects instructed not to rate the mouthfeel of oleic acid, but only the chemosensory perception?
  3. Electrophysiology: was the baseline signal at similar levels between taster groups? If not, please indicate how the signal was normalized.
  4. Density assessments of fungiform taste papillae: were all papillae counted or only the papillae that had a visible taste pore?
  5. Figure 1 vs Figure 2 vs Figure 3: Signal amplitude for oleic acid looks very different as well as the timing for signal increase. Please clarify/discuss.
  6. Figure 2: X axis numbers with decimals should use a period instead of a comma.
  7. Discussion: it would be interesting to discuss further why some studies do not report any correlation between PROP status, taste sensitivity and papilla density, compared to the present study and others.

Author Response

We have reworked the manuscript according to comments of the Reviewers.

In the revised manuscript the changes made according to Reviewer 1 are highlighted in red and those according to Reviewer 2 in blue.

 Comments to the Author:

Although multiple articles have investigated the relationship between PROP taster status, fungiform papilla density and taste sensitivity in humans, the existence of a correlation remains controversial. The article by Melis and colleagues presents new data that support a correlation between PROP taster status, fungiform papilla density and taste sensitivity. While the study could have been even stronger with a larger sample size, the data presented here are convincing as the methodologies are very thorough and include two different measures of taste sensitivity: taste intensity rating by the subjects on a Labelled Magnitude Scale, and unbiased electrophysiological recording. In addition to the five basic taste modalities, the authors also included fat taste sensitivity, which has been proposed to be the sixth basic taste, for a broader picture. The article is well written and includes very detailed procedures and statistical analysis.

Reply: We appreciate the Reviewer's nice words about our paper.

Only a few minor comments arose from reading the manuscript:

  1. Line 186: what was the oleic acid concentration? What was the vehicle?

Reply: Oleic acid stimulation was delivered by placing on the circular area of the tongue surface a paper disk which was impregnated with 30 µl of undiluted oleic acid. We recognize that this point was not accurately described, and this generated confusion. We modified the text accordingly at line 194.

  1. Line 193: dry paper disks were used as controls, but it is unclear how the generated control data were included in the analysis. Please clarify. Also, were some paper disks impregnated with water only, or with a texture agent only, as vehicle control (for sweet, bitter, sour, umami, salty compounds) and control for the texture/mouthfeel of oleic acid, respectively? Were the subjects instructed not to rate the mouthfeel of oleic acid, but only the chemosensory perception?

Reply: We clarified at lines 199-200 that, in order to verify that the potential variations recorded were certainly due to taste stimulations and not to mechanical stimulations caused by placing the disks, subjects were also tested with dry paper disks. In addition, in the Results section we explained that stimulations with dry paper disks produced no potential changes (line 232). Consequently, they were not included in the analysis.

We have not used disks impregnated with water (the only vehicle used), but we think that if the electrophysiological signals recorded were due to water, each taste quality would not have evoked a voltage change with a characteristic time course during stimulation.

Yes, the subjects were instructed to rate only the chemosensory perception. We specified this at line 138-139.

  1. Electrophysiology: was the baseline signal at similar levels between taster groups? If not, please indicate how the signal was normalized.

Reply: Yes, the baseline signal was at similar levels between taster groups. In any case, the voltage changes (amplitude values) in response to six taste qualities were obtained by subtracting the baseline value from the voltage at 0.1, 2.5, 5, 10 and 15 s from stimulation onset. We better explained this point at lines 185-187

  1. Density assessments of fungiform taste papillae: were all papillae counted or only the papillae that had a visible taste pore?

Reply: we counted the total number of papillae according to Melis et al. 2013 (92). We better explained this point at line 202-204.

 5. Figure 1 vs Figure 2 vs Figure 3: Signal amplitude for oleic acid looks very different as well as the timing for signal increase. Please clarify/discuss.

Reply: We comply with the Reviewer’s request. We modified the text, at line 352-357, as follows: “Finally, we recorded the lowest and slowest hyperpolarization in the responses to oleic acid in which the signal amplitude reached values comparable to those of other stimuli only at the end of stimulation. This slow electrophysiological activation in response to oleic acid, that has been already observed in our previous work [81], could depend on high surface tension of this molecule that determines its slow diffusion towards the cell that detects it”.

 6. Figure 2: X axis numbers with decimals should use a period instead of a comma.

Reply: we modified X axis numbers in figure 1 accordingly.

 7. Discussion: it would be interesting to discuss further why some studies do not report any correlation between PROP status, taste sensitivity and papilla density, compared to the present study and others.

Reply: we provide a brief discussion of this issue on lines 398-404.

Reviewer 2 Report

Melis et al., test the ability of electrical signals to classify taster status of individuals. Previous work has suggested those with super senstivity to PROP correlates with papillae density and taster status to other substances, however being a psychophysical analysis subjectivity can influence results. The method reported here would permit classification and testing on a non-subjective technique. The results overall are clearly presented. The authors show that super-taster have significantly different electrical responses to chemicals that can be distinguished from medium and/or non-tasters. Overall the manuscript is well written with only some minor issues.

1) Figure showing the average and representative traces from each subject/each tastant would be more informative for figure 1. A schematic of the recording setup would also be beneficial.

2) Figure 2: To be consistent with text, x-axis should be labeled with 0.1 and 2.5 instead of 0,1 and 2,5.

3) The size of symbols on graphs obscures error bars for several figures and the letters used to denote significance can be hard to read on a few.

4) representative images of Super, medium adn non taster for tongues would benefit figure 6

Author Response

Comments to the Author:

Melis et al., test the ability of electrical signals to classify taster status of individuals. Previous work has suggested those with super senstivity to PROP correlates with papillae density and taster status to other substances, however being a psychophysical analysis subjectivity can influence results. The method reported here would permit classification and testing on a non-subjective technique. The results overall are clearly presented. The authors show that super-taster have significantly different electrical responses to chemicals that can be distinguished from medium and/or non-tasters. Overall the manuscript is well written with only some minor issues.

Reply: We appreciate the Reviewer's nice words about our paper.

1) Figure showing the average and representative traces from each subject/each tastant would be more informative for figure 1. A schematic of the recording setup would also be beneficial.

 Reply: we are not sure what the Reviewer is asking for, but if he is asking to show representative traces of subjects belonging to different PROP taster categories, we would like to emphasize that we presented figure 1 (now figure2) in order to show the differences among the waveforms of the potential changes recorded in response to different taste stimuli, not differences related to PROP taster status. Accordingly, we are sorry, but we are very convinced that the figure should not be changed.

Differently, we agree with the Reviewer that a scheme of the recording setup could be beneficial. In this respect, we added a figure in the Method section.

2) Figure 2: To be consistent with text, x-axis should be labeled with 0.1 and 2.5 instead of 0,1 and 2,5.

Reply: we modified X axis numbers in figure 1 accordingly.

3) The size of symbols on graphs obscures error bars for several figures and the letters used to denote significance can be hard to read on a few.

Reply: We comply with the Reviewer’s request. The size of symbols on graphs of Figure 2 (now figure 3) have been changed.

4) representative images of Super, medium and non-taster for tongues would benefit figure 6.

Reply: We comply with the Reviewer’s request. In the figure 6 (now figure 7), we included images of the stained area (6 mm dia) of the tongue where the fungiform papillae were counted in a representative super-taster, medium taster and non-taster.